# Nanosecond-time-scale delayed fluorescence molecule for deep-blue OLEDs with small efficiency rolloff

Jong Uk Kim[1,2], In Seob Park[1], Chin-Yiu Chan[1], Masaki Tanaka [1], Youichi Tsuchiya [1], Hajime Nakanotani [1,2,3] & Chihaya Adachi [1,2,3 ✉]

Aromatic organic deep-blue emitters that exhibit thermally activated delayed fluorescence (TADF) can harvest all excitons in electrically generated singlets and triplets as light emission. However, blue TADF emitters generally have long exciton lifetimes, leading to severe efficiency decrease, i.e., rolloff, at high current density and luminance by exciton annihilations in organic light-emitting diodes (OLEDs). Here, we report a deep-blue TADF emitter employing simple molecular design, in which an activation energy as well as spin–orbit coupling between excited states with different spin multiplicities, were simultaneously controlled. An extremely fast exciton lifetime of 750 ns was realized in a donor–acceptor-type molecular structure without heavy metal elements. An OLED utilizing this TADF emitter displayed deep-blue electroluminescence (EL) with CIE chromaticity coordinates of (0.14, 0.18) and a high maximum EL quantum efficiency of 20.7%. Further, the high maximum efficiency were retained to be 20.2% and 17.4% even at high luminance.

[1] Center for Organic Photonics and Electronics Research (OPERA) and Department of Applied Chemistry, Kyushu University, 744 Motooka, Nishi-ku, Fukuoka 819-0395, Japan. [2] JST, ERATO, Adachi Molecular Exciton Engineering Project, Kyushu University, 744 Motooka, Nishi-ku, Fukuoka 819-0395, Japan. [3] International Institute for Carbon Neutral Energy Research (WPI-I2CNER), Kyushu University, 744 Motooka, Nishi-ku, Fukuoka 819-0395, Japan. ✉email: adachi@cstf.kyushu-u.ac.jp

Organic light-emitting diodes (OLEDs) have been commercialized in flat panel displays and solid-state lighting applications, and significant efforts are still devoted to enhancing OLED performance. In OLEDs, the most important parameter is the internal quantum efficiency ($\eta_{int}$), which is theoretically limited to 25% in traditional fluorescence-based OLEDs, as only singlet excitons can be harvested under electrical excitation[1,2]. On the other hand, the utilization of phosphorescent emitters containing heavy metals such as Ir, Pt, Os, and Au enhanced intersystem crossing by the strong spin–orbit coupling (SOC), and these phosphorescent emitters can harvest not only singlet excitons but also triplet excitons, leading to an ideal $\eta_{int}$ of nearly 100% in OLEDs[3,4]. However, the weak metal–ligand coordination bonds result in a limited device lifetime in blue OLEDs[5–7]. As an alternative approach, highly efficient thermally activated delayed fluorescence (TADF)-based OLEDs have recently been realized using simple aromatic compounds as an emitter[8]. In this system, triplet excitons are efficiently upconverted from a lowest triplet state ($T_1$) to the lowest excited singlet state ($S_1$) by a reverse intersystem crossing (RISC) process, governed by a small energy gap ($\Delta E_{ST}$) between the $S_1$ and $T_1$ states, resulting in a maximum $\eta_{int}$ of close to 100%. Eventually, high external quantum efficiencies ($\eta_{ext}$) of over 20% have been achieved for TADF-OLEDs[9–31].

Nevertheless, blue TADF-OLEDs suffer from severe efficiency rolloff compared to their green and red counterparts because relatively long-lived triplet excitons in blue TADF molecules directly affect the operational stability and efficiency rolloff characteristics of TADF-OLEDs. These effects are observed because of the increase of exciton deactivation processes at high current density, including triplet–triplet annihilation (TTA) and singlet–triplet annihilation (STA)[32,33]. Advanced blue TADF molecules with ideally short exciton lifetimes (<1 μs) are thus essentially required for future OLED applications. To realize short exciton lifetimes in pure organic TADF molecules, a rate constant of RISC ($k_{RISC}$, $T_1 \rightarrow S_1$) is the most critical parameter because TADF molecules emit light primarily via the $S_1 \rightarrow T_1 \rightarrow S_1 \rightarrow S_0$ and $T_1 \rightarrow S_1 \rightarrow S_0$ delayed processes with the $S_1 \rightarrow S_0$ prompt process under electrical excitation. According to the first-order perturbation theory, that is, Fermi's golden rule, $k_{RISC}$ between the two states is proportional to $\langle S|\hat{H}_{SOC}|T\rangle/\Delta E_{ST}$[34–36]:

$$k_{RISC} \propto \left|\langle S|\hat{H}_{SOC}|T\rangle\right|^2 \exp\left(\frac{-\Delta E_{ST}}{k_B T}\right). \quad (1)$$

Here, $\langle S|\hat{H}_{SOC}|T\rangle$ is the SOC matrix element between the excited

singlet (S) and triplet (T) states, $k_B$ is the Boltzmann constant, and $T$ is temperature. Using this relationship, minimization of $\Delta E_{ST}$ is a widely adopted strategy to achieve efficient intramolecular charge-transfer (CT) of TADF molecules[8,11–30,32]. However, the spin-flip processes, that is, intersystem crossing ISC and RISC, between excited CT singlet ($^1CT$) and triplet ($^3CT$) states are very inefficient according to the El-Sayed rule because of the independent electric dipole moment with an electron spin, resulting from a weak SOC matrix element[37,38]. In contrast, remarkably strong SOC can be expected when the spin-flip processes arise between the CT and energetically close-lying locally excited (LE) states with different spin multiplicities owing to the orbital angular momentum change between the two states[27,39]. Several groups have reported that $k_{RISC}$ is accelerated by the strong SOC matrix element between the $^1CT$ ($S_1$) and $^3LE$ ($T_2$) states, caused by non-adiabatic vibronic coupling ($\hat{H}_{VC}$) between the $^3CT$ ($T_1$) and $^3LE$ ($T_2$) states[40–42]. In this study, we demonstrate that a small modulation in excited states of an aromatic organic molecule, mediated by SOC between $^1CT$ and $^3LE$ states, greatly affects the rate constants of a TADF molecule. As a result, we attained an ideal nano-second-scale exciton lifetime of 750 ns in a deep-blue TADF molecule, which can drastically minimize efficiency rolloff in TADF-OLEDs.

## Results

**Molecular design and synthesis**. To validate our concept, we designed a donor–acceptor (D–A)-type TADF molecule, **TMCz-BO** (Fig. 1), by introducing 1,3,6,8-tetramethyl-9H-carbazole (**TMCz**) as a donor unit and 5,9-dioxa-13b-boranaphtho[3,2,1-de]anthracene (**BO**) as an acceptor unit. Both units have ideally high $T_1$ energies (**TMCz** and **BO**[43]: $E_T = 3.00$ eV, Supplementary Fig. 1), which are similar to $^1CT$ of previously reported D–A-type deep-blue TADF emitters[14,16,17,21–24,28–30,32]; therefore, we expected to attain ideal $^1CT \approx {}^3LE$. Moreover, the small spatial overlap of this molecule between the HOMO (highest occupied molecular orbital) and LUMO (lowest unoccupied molecular orbital) in **TMCz** and **BO**, respectively, induced the CT character in both $S_1$ and $T_1$ states, resulting in a small calculated $\Delta E_{ST}$ of 0.01 eV (i.e., $^1CT \approx {}^3CT$). This analysis was performed using time-dependent density functional theory (TDDFT) at the PBE0/6-31G(d) level in the gas phase (Supplementary Table 1). To control the $^3LE$ state of an acceptor unit, we also designed a model molecule, **TMCz-3P**, consisting of 3,11-diphenyl-5,9-dioxa-13b-boranaphtho[3,2,1-de]anthracene (**3P**) as an acceptor,

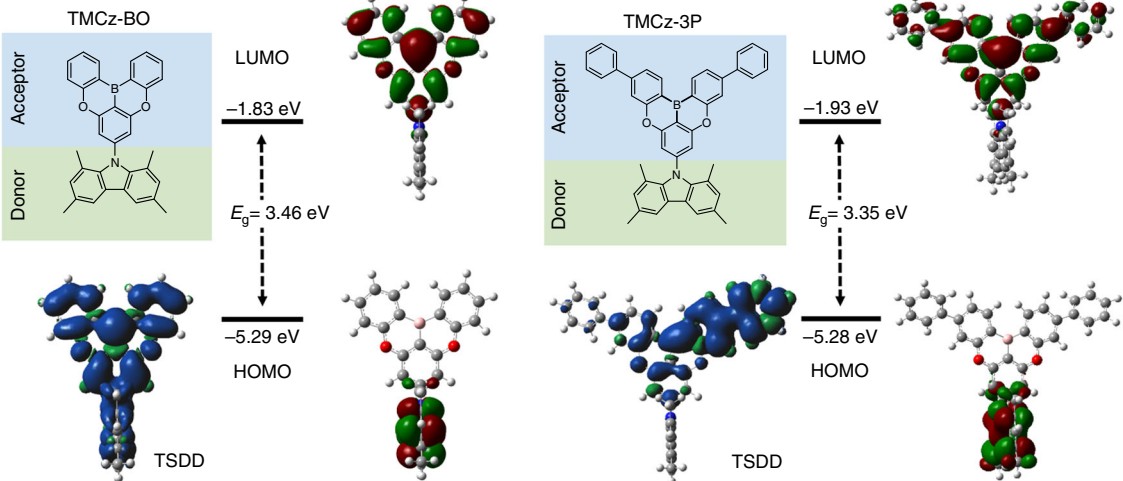

**Fig. 1 Molecular structures and energy levels.** HOMO and LUMO of **TMCz-BO** and **TMCz-3P** characterized by DFT calculations at the PBE0/6-31G(d) level of theory and TSDDs in $T_1$ state.

which had a lower $E_T$ (2.76 eV) than that of **BO** because additional phenylenes lead to longer π-conjugation, thereby decreasing $E_T$. Although this molecule has a similarly small frontier orbital overlap, relatively strong LE character was observed rather than the CT character in its $T_1$ state according to triplet spin density distribution (TSDD) simulations using the optimized ground-state geometry. In addition, TSDD simulations revealed that **TMCz-BO** has a strong CT rather than LE character in its $T_1$ state because the TSDD of **TMCz-BO** resided throughout the entire D–A molecule, whereas TSDD was only displayed in **TMCz-3P** on its acceptor unit (**3P**, Fig. 1).

Figure 2 shows the synthetic routes used to produce the boron-based compounds. The intermediates **3** and **4** were prepared from **1** and **2** using the Buchwald–Hartwig amination with high yields of 80%. The final compounds were obtained using the cyclization reactions of **3** (for **TMCz-BO**) or **4** (for **TMCz-3P**) in the presence of n-butyllithium (n-BuLi) and boron tribromide (BBr₃). These final products were purified using temperature-gradient vacuum sublimation. The chemical structures of these compounds were characterized using ¹H and ¹³C nuclear magnetic resonance (NMR) spectroscopy, mass spectrometry (MS), and elemental analysis. The detailed synthetic procedures and characterization data are provided in the Methods section and Supplementary Methods.

**Photophysical properties**. The fundamental photophysical properties of **TMCz-BO** and **TMCz-3P** were first studied in toluene solution at a concentration of $10^{-5}$ M (Table 1 and Supplementary Figs. 2, 3). As shown in Fig. 3a, two clear absorption bands in the ultraviolet–visible (UV–vis) absorption spectra were observed for both materials. The shorter wavelength is attributed to the π–π* transition of the acceptor and donor units, whereas the other one at longer wavelength (>370 nm) corresponds to an intramolecular CT transition from the donor to acceptor units. To attain a deeper understanding of their excited-state properties, we also investigated the solvatochromic effects of **TMCz-BO** and **TMCz-3P** in various solvents (Supplementary Fig. 2). Fluorescence spectra with large bathochromic shifts were observed when the solvent polarity was changed from non-polar cyclohexane to polar dichloromethane. The maximum peak wavelength ($\lambda_{PL}$) of **TMCz-BO** was ~407 nm in cyclohexane and 503 nm in dichloromethane, whereas $\lambda_{PL}$ of **TMCz-3P** was 416 nm in cyclohexane and 517 nm in dichloromethane. Owing to increased solvent polarity, there was a large difference in the emission peak wavelength ($\Delta\lambda_{max} = 96$ nm for **TMCz-BO** and $\Delta\lambda_{max} = 101$ nm for **TMCz-3P**), indicating pronounced positive solvatochromism, which confirmed the strong CT character of **TMCz-BO** and **TMCz-3P** in their $S_1$ states.

For further investigation of the photophysical and TADF properties, doped films of both emitters in an amorphous host matrix (i.e., PPF: 2,8-bis(diphenylphosphoryl)dibenzo[b,d]furan, $E_T = 3.1$ eV)[44] were prepared to avoid concentration quenching. Here, we selected PPF as a host since it can provide both efficient triplet confinement and electron transport abilities. As observed in Fig. 3, the 30 wt% doped films of **TMCz-BO** and **TMCz-3P** exhibited blue emission with peaks at $\lambda_{PL} = 467$ and 477 nm,

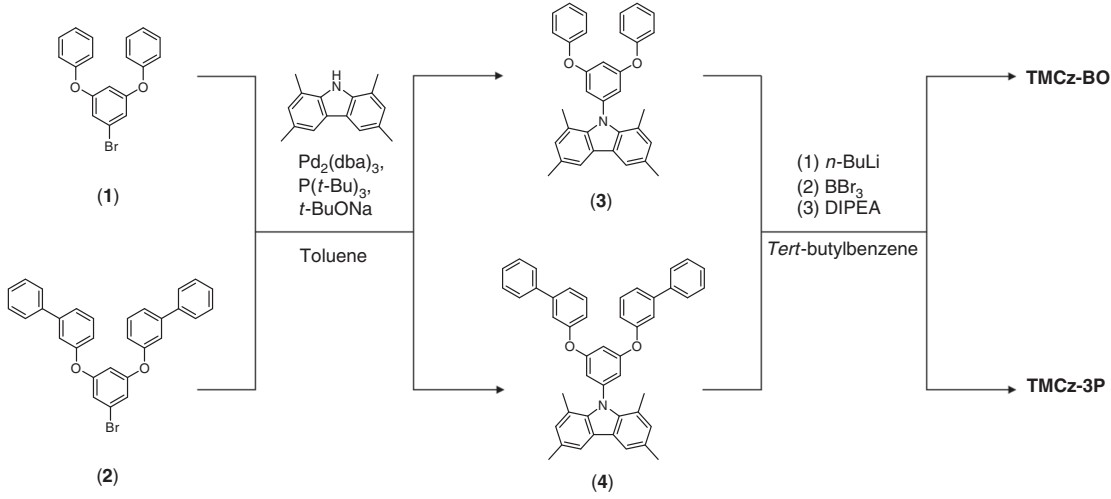

**Fig. 2 Synthetic scheme.** Synthetic routes for **TMCz-BO** and **TMCz-3P**.

**Table 1 Photophysical characteristics of TMCz-BO and TMCz-3P.**

| Compound | $\lambda_{abs}$ (nm) Sol[a] | $\lambda_{PL}$ (nm) Sol[a]/film[b] | $\Phi_{PL}$ (%)[c] Sol[a]/film[b] | $\tau_p$ (ns)[d]/$\tau_d$ (μs)[d] | HOMO (eV)[e] | LUMO (eV)[f] | $E_S/E_T$ (eV)[g] | $\Delta E_{ST}$ (meV)[h] | $E_a$ (meV)[i] |
|---|---|---|---|---|---|---|---|---|---|
| **TMCz-BO** | 282,377 | 446/467 | 81/98 | 38/0.75 | −5.93 | −2.77 | 2.95/2.93 | 20.0 | 13.4 |
| **TMCz-3P** | 298,386 | 455/477 | 56/76 | 29/14.5 | −5.97 | −2.89 | 2.88/2.74 | 134 | 39.8 |

[a]Measured in oxygen-free toluene at room temperature (298 K).
[b]30 wt% doped thin film in a host matrix (host = PPF).
[c]Absolute PLQY evaluated using an integrating sphere under a nitrogen atmosphere.
[d]PL lifetimes of prompt ($\tau_p$) and delayed ($\tau_d$) decay components for 30 wt% doped film at 298 K.
[e]Determined by photoelectron yield spectroscopy in pure neat films.
[f]Deduced from the HOMO and optical energy gap ($E_g$).
[g]Singlet ($E_S$) and triplet ($E_T$) energies estimated from onsets of the emission spectra at 298 and 77 K in 30 wt% doped films, respectively.
[h]$\Delta E_{ST} = E_S - E_T$.
[i]Activation energies of RISC in 30 wt% doped films.

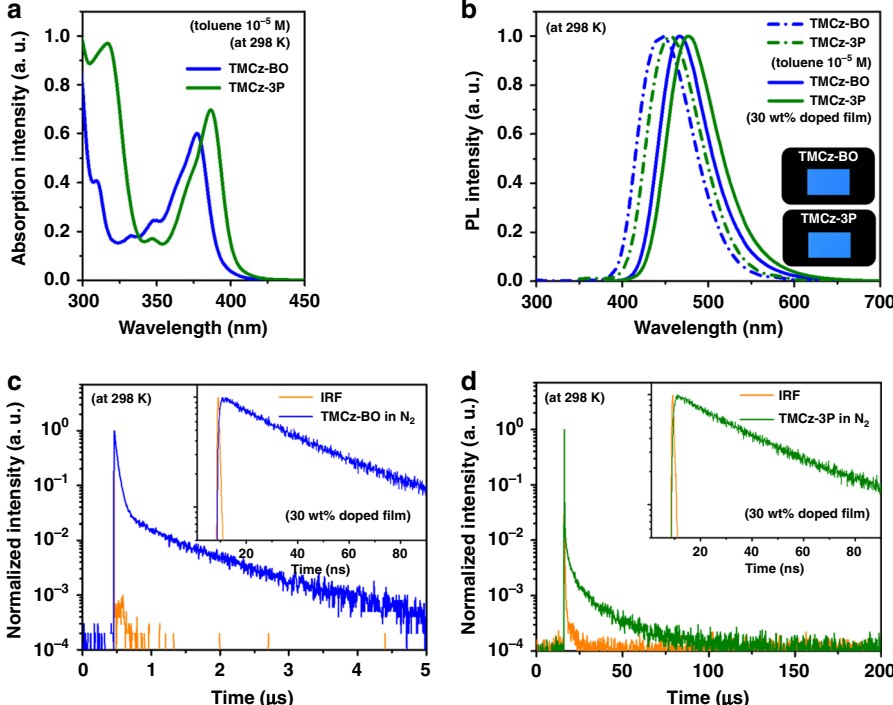

**Fig. 3 Absorption spectra and PL transient decay. a** Absorption spectra of **TMCz-BO** and **TMCz-3P** in $10^{-5}$ M toluene. **b** PL emission spectra of **TMCz-BO** and **TMCz-3P** in $10^{-5}$ M toluene and 30 wt% doped films in a PPF host matrix. Transient PL decay curves of **c** 30 wt% **TMCz-BO** [inset, $\tau_p$ (ns)] and **d TMCz-3P** [inset, $\tau_p$ (ns)] doped films in a PPF host matrix.

corresponding to Commission Internationale de l'Éclairage (CIE) chromaticity coordinates of (0.15, 0.18) and (0.16, 0.25), respectively. We note that the red-shifted emission maxima of the doped films were observed compared to those of their toluene solution ($\lambda_{PL} = 446$ nm for **TMCz-BO** and $\lambda_{PL} = 455$ nm for **TMCz-3P**), which should be primarily associated with a stronger dipole moment of PPF (5.8 D) than that of toluene (0.36 D)[27]. Such a phenomenon is an additional evidence that these molecules possess a strong CT character.

From the transient PL characteristics of the doped films at 300 K, we observed a clear nano-second-scale prompt component and a microsecond-scale delayed PL component, which were fitted using the biexponential model. As shown in Fig. 3c, **TMCz-BO** has an extremely fast delayed emission lifetime ($\tau_d$) of 750 ns (fractional delayed PL quantum yield: $\Phi_d = 32\%$) with a prompt emission lifetime ($\tau_p$) of 38 ns (fractional prompt PL quantum yield: $\Phi_p = 66\%$) in $N_2$, whereas a relatively long $\tau_d$ of 14.5 µs ($\Phi_d = 11\%$) with similar $\tau_p$ of 29 ns ($\Phi_p = 65\%$) was observed in the **TMCz-3P**-based doped film (Fig. 3d). The rate constants, that is, $k_r$, $k_{ISC}$, and $k_{RISC}$, were also estimated for both doped films using the reported method and summarized in Supplementary Table 3[27]. Although similar $k_r$ and $k_{ISC}$ values were determined from both doped films (**TMCz-BO**: $k_r = 1.7 \times 10^7\,s^{-1}$; $k_{ISC} = 0.9 \times 10^7\,s^{-1}$ and **TMCz-3P**: $k_r = 2.3 \times 10^7\,s^{-1}$; $k_{ISC} = 1.2 \times 10^7\,s^{-1}$), **TMCz-BO** has a large $k_{RISC}$ of $1.9 \times 10^6\,s^{-1}$, which is two orders of magnitude higher than that of **TMCz-3P** ($3.3 \times 10^4\,s^{-1}$).

To understand the exciton dynamics in the excited states of **TMCz-BO** and **TMCz-3P**, we first measured the phosphorescence (77 K) spectra of 5–30 wt% doped films in the PPF host matrix with fluorescence (298 K) to determine whether they exhibited CT or LE character in their $T_1$ states. The phosphorescence spectra of 5–30 wt% **TMCz-BO**-doped films contained broad and structureless characteristics (i.e., CT character) without the vibrational mode, similar to the $^3$LE of both the donor (**TMCz**) and acceptor (**3P**) units (Fig. 4a). Furthermore,

$E_T$ (2.93 eV) of **TMCz-BO**, which was estimated from the onset of the phosphoresce spectrum of the 30 wt% doped film, was quite close to those of both **TMCz** and **BO** ($^3$LE = 3.00 eV). In contrast, **TMCz-3P** was observed to have a similar $E_T$ (2.76 eV) as that of the acceptor (**3 P**, $^3$LE = 2.76 eV) (Fig. 4b). The phosphorescence spectra of **TMCz-3P** coincided well with that of the acceptor (**3P**) unit with appreciable redshift to the onset of the fluorescence. In addition, LE character with vibrational mode was observed in the phosphorescence spectra of 5–20 wt% **TMCz-3P**-based doped films, even though the phosphorescence spectrum appeared to indicate CT character in the 30 wt% doped film. As a next step, we also analyzed the temperature dependences of $k_{RISC}$ (Fig. 4c) using the 30 wt% doped films. According to the classical Arrhenius equation, $k_{RISC}$ is given as $k_{RISC} = A\exp(-E_a/k_BT)$, where $A$ is the frequency factor involving the SOC constant. From the Arrhenius plots of $k_{RISC}$, similarly small $E_a$ values were experimentally estimated to be 13.4 meV for **TMCz-BO** and 39.8 meV for **TMCz-3P**. Moreover, small $E_a$ values for 5–20 wt% doped films were also analyzed in the range of 13.7–14.5 meV for **TMCz-BO** and 24.0–34.0 meV for **TMCz-3P** (Supplementary Fig. 4 and Table 2). These results indicated that the large difference in $k_{RISC}$ values between **TMCz-BO** and **TMCz-3P** is strongly associated with SOC in contrast with in common TADF systems.

To obtain additional insight into the spin-flip RISC process, we carefully analyzed the relationship between the SOC and $E_a$ values with consideration of their $\Delta E_{ST}$ values. Although **TMCz-BO** has a similarly small $\Delta E_{ST}$ of 20 meV with its $E_a$, $\Delta E_{ST}$ of **TMCz-3P** was estimated to be quite large (134 meV), which is three times larger than its $E_a$. These experimentally obtained $k_{RISC}$, $E_a$, and $\Delta E_{ST}$ suggest that different spin-flip processes should be involved in **TMCz-BO** and **TMCz-3P**. As suggested by the energy level diagram in Fig. 5a, the efficient RISC spin-flip process for **TMCz-BO** should be involved in $^3$CT → $^3$LE → $^1$CT because energetically close-lying excited states induce efficient $\hat{H}_{VC}$ from the $^3$CT

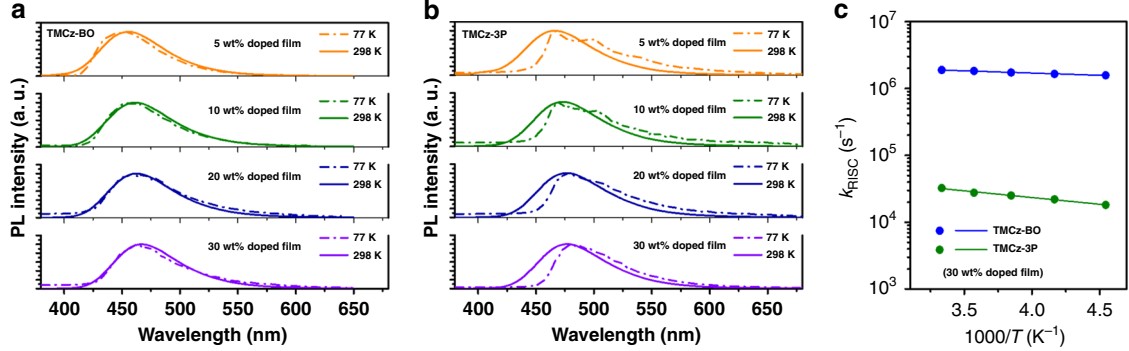

**Fig. 4 PL spectra at room temperature and 77 K. a**, **b** Fluorescence (line, 298 K) and phosphorescence (dash dot, 77 K) spectra of doped films for **a TMCz-BO** and **b TMCz-3P**. **c** Arrhenius plots of rate constants of RISC ($k_{RISC}$) obtained with the doped films of 30 wt% **TMCz-BO** and **TMCz-3P**.

**Table 2 EL performance of blue TADF-OLEDs.**

| TADF emitter | $\lambda_{EL}$ (nm) | $\lambda_{FWHM}$ (nm) | $E_{FWHM}$ (eV) | $V_{on}$ (V) | $L_{max}$ (cd m$^{-2}$) | $\eta_{ext}$ (%) Max/@100 cd m$^{-2}$/ @1000 cd m$^{-2}$ | $\eta_c$ (cd A$^{-1}$) | $\eta_p$ (lm W$^{-1}$) | CIE (x, y) |
|---|---|---|---|---|---|---|---|---|---|
| TMCz-BO | 471 | 59 | 0.327 | 3.0 | 5900 | 20.7/20.2/17.4 | 29.8 | 31.2 | (0.14, 0.18) |
| TMCz-3P | 479 | 61 | 0.324 | 2.8 | 6500 | 20.4/18.3/12.8 | 37.8 | 40.3 | (0.14, 0.26) |

$\lambda_{EL}$ EL emission maximum, $\lambda_{FWHM}$ and $E_{FWHM}$ full-width at half-maximum, $V_{on}$ turn-on voltage at 1 cd m$^{-2}$, $L_{max}$ maximum luminance, $\eta_{ext}$ external EL quantum efficiency, $\eta_c$ maximum current efficiency, $\eta_p$ maximum power efficiency, CIE (x, y) Commission Internationale de l'Éclairage color chromaticity coordinates measured at 10 mA cm$^{-2}$.

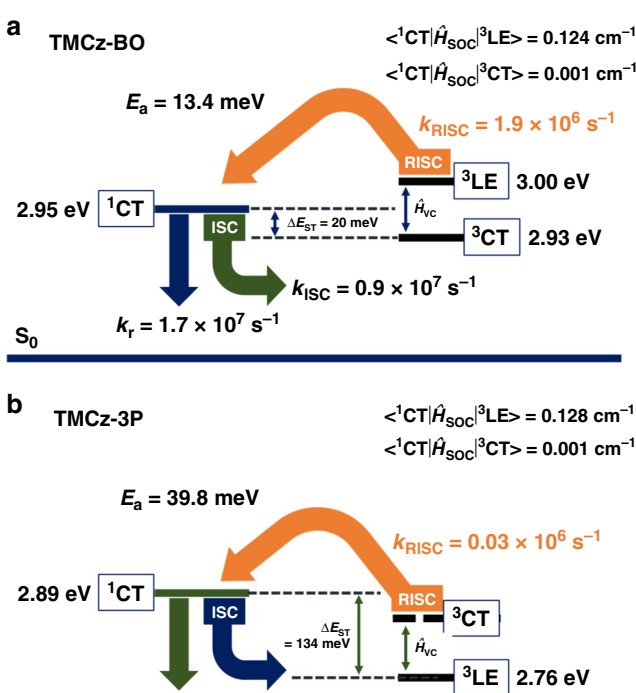

**Fig. 5 Schematic illustration of plausible TADF mechanisms. a** Decay and upconversion processes in **TMCz-BO** and **b TMCz-3P**. The energy levels and rate constants were estimated using 30 wt% doped films.

to $^3$LE states, and then RISC can be accelerated from $^3$LE to $^1$CT states mediated by a much larger SOC matrix element ($<^1$CT| $\hat{H}_{SOC}|^3$LE> = 0.124 cm$^{-1}$) than $<^1$CT|$\hat{H}_{SOC}|^3$CT> = 0.001 cm$^{-1}$, resulting in its extraordinarily fast emission lifetime of 750 ns. In addition, it is difficult for the spin-flip process in the **TMCz-3P** molecule to occur along the same pathway as that for

**TMCz-BO**, even though it has a similarly large $<^1$CT|$\hat{H}_{SOC}|$ $^3$LE> = 0.128 cm$^{-1}$, because its experimental $\Delta E_{ST}$ is much larger than the thermal energy ($k_B T \approx 25.9$ meV) at 300 K as well as its $E_a$ (Fig. 5b). Experimental data thus suggest that the $^3$CT → $^1$CT spin-flip process (i.e., hyperfine coupling)[45,46] should occur after efficient $\hat{H}_{VC}$ from $^3$LE to energetically close-lying higher $^3$CT states, leading to relatively small $k_{RISC}$ because of the negligible $<^1$CT|$\hat{H}_{SOC}|^3$CT> of 0.001 cm$^{-1}$. Therefore, energetically close-lying excited states (i.e., $^1$CT ≈ $^3$CT ≈ $^3$LE) with suitably large SOC between $^1$CT and $^3$LE should be realized to ensure extremely fast emission lifetime in TADF molecules.

**Device characterization and performance.** Employing **TMCz-BO** and **TMCz-3P** as emitters, two multi-layered OLEDs were fabricated using the following device architecture: indium tin oxide (ITO, 50 nm)/HAT-CN (2,3,6,7,10,11-hexacyano-1,4,5,8,9,12-hexaazatripheny) (10 nm)/TAPC (4,4′-cyclohexylidenebis[N,N-bis(4-methylphenyl)benzenamine]) (50 nm)/CCP (9-phenyl-3,9′-bicarbazole) (10 nm)/EML (20 nm)/PPF (10 nm)/B3PyPB (1,3-bis[3,5-di(pyridine-3-yl)phenyl] benzene) (30 nm)/Liq (8-hydroxyquinoline lithium) (1 nm)/Al (100 nm), in which HAT-CN and Liq were used as hole injection and electron injection layers, respectively. The materials TAPC and B3PyPB were used as hole transport and electron transport layers, respectively. Thin layers of CCP and PPF with a high $T_1$ energy (3.0 and 3.1 eV) were inserted to suppress triplet exciton quenching at the neighboring interfaces and to confine the excitons inside the emitting layers[14,16,17,21–24,28–30,32,47].

HOMO, LUMO, and thermal properties of **TMCz-BO** and **TMCz-3P** are summarized in Supplementary Figs. 5–7a. The EL characteristics of both devices are depicted in Fig. 6, and the key device parameters are summarized in Table 2. As shown in Fig. 6a, the devices based on **TMCz-BO** and **TMCz-3P** exhibited blue EL emission with peaks ($\lambda_{EL}$) at 471 nm (deep blue) and 479 nm (blue) with corresponding CIE chromaticity coordinates of (0.14, 0.18) and (0.14, 0.26), respectively, which were consistent with their corresponding PL spectra. It is noteworthy that the

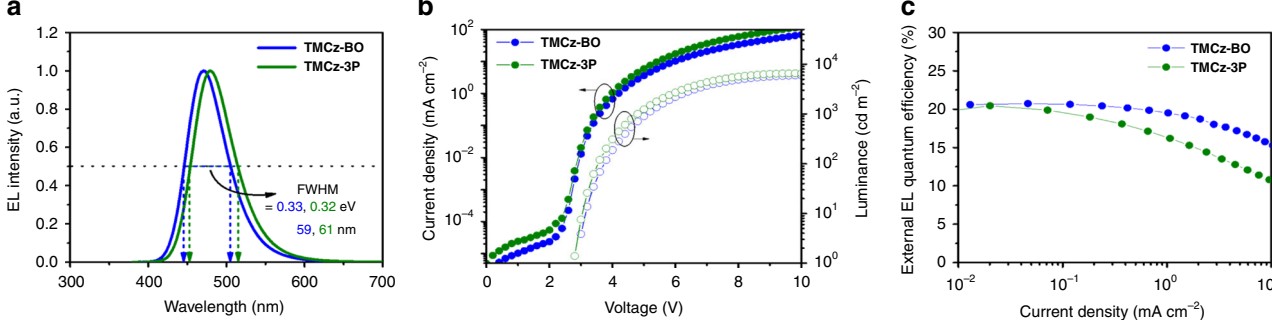

**Fig. 6 OLED characteristics. a** Normalized EL spectra, **b** current density–voltage–luminance (J–V–L) characteristics, and **c** external EL quantum efficiency ($\eta_{ext}$) versus luminance plots of **TMCz-BO-** and **TMCz-3P**-based OLEDs.

devices displayed a rather narrow full-width at half-maximum (FWHM, $\lambda_{FWHM}$) of 59 nm ($E_{FWHM} = 0.327$ eV) for **TMCz-BO** and 61 nm ($E_{FWHM} = 0.324$ eV) for **TMCz-3P**, resulting from minimizing molecular vibrations using the rigid acceptor structure and minimizing molecular conformation changes in their ground and $S_1$ states using the *peri*-position dimethyl groups in TMCz[27,31,47].

Figure 6b, c shows the current density–voltage–luminance (J–V–L) and external EL quantum efficiency versus current density ($\eta_{ext}$–J) plots. Both devices exhibited rather low turn-on voltages ($V_{on}$) in the range of 2.8–3.0 V and achieved high maximum $\eta_{ext}$ exceeding 20% (20.7% for **TMCz-BO** and 20.4% for **TMCz-3P**). This high efficiency of the **TMCz-3P**-based device drastically decreased with increasing current density and luminance, which is similar to the trend observed for most reported blue TADF-OLEDs. As a result, low $\eta_{ext}$ values (18.3% at 100 cd m$^{-2}$ and 12.8% at 1000 cd m$^{-2}$) were obtained from the **TMCz-3P**-based device. However, the **TMCz-BO**-based device retained high $\eta_{ext}$ values of 20.2% at 100 cd m$^{-2}$ and 17.4% at 1000 cd m$^{-2}$, corresponding to a 2.4% and 15.9% decrease in $\eta_{ext}$, respectively. Low-efficiency rolloff is primarily attributed to its nano-second-order emission lifetime, which efficiently suppresses the accumulation of triplet excitons within **TMCz-BO**; hence, TTA and/or STA could be minimized in the device. To the best of our knowledge, the efficiency decrease ratio of the **TMCz-BO**-based device is the small of those reported for TADF molecules for deep-blue OLEDs (CIEy ≤ 0.20) to date (Supplementary Table 4)[14,16,17,21–24,28–30,32,47]. We note that the recently some sophisticated devices showed rather relaxed rolloff behavior even with the TADF emitters with microsecond-order delayed lifetimes. Probably wide recombination and exciton formation width may contribute it. Thus, it should be considered to improve total device performance from the aspect of comprehensive control of delayed lifetime, charge carrier transport, and recombination characteristics[48].

## Discussion

Kaji and co-workers[49] recently reported a through-space CT molecule, TpAT-tFFO, by introducing face-to-face alignment of the donor and acceptor units (Fig. 7a). Such a through-space CT character in a single molecule induces extremely small spatial overlap between the HOMO and LUMO, resulting in a very small $\Delta E_{ST}$. Although an exceedingly large $k_{RISC}$ value of over $10^7$ s$^{-1}$ was achieved in the doped film, an unusually slow rate constant of fluorescence radiative decay ($k_r = 1.1 \times 10^6$ s$^{-1}$) was observed. This slow rate constant can be attributed to the negligible spatial overlap of the through-space molecule, leading to a decrease of the transition dipole moment, which should affect its small radiative decay, according to Fermi's golden rule. As a result, the doped film exhibited a relatively long delayed emission lifetime ($\tau_d$) of 4.1 μs. In 2018, Yasuda and co-workers[27] successfully

designed a non-metallic TADF molecule, MPAc-BS, with the largely enhanced SOC matrix element of <$^1$CT|$\hat{H}_{SOC}$|$^3$LE> = 4.67 cm$^{-1}$ (in general, aromatic organic molecules have very small <$^1$CT|$\hat{H}_{SOC}$|$^3$LE> ≤ 0.10 cm$^{-1}$), thereby attaining a large $k_{RISC}$ of $3.5 \times 10^6$ s$^{-1}$ (Fig. 7b). However, this molecule also possessed a very large rate constant of ISC ($k_{ISC} = 9.9 \times 10^7$ s$^{-1}$), which is over 10 times larger than that of $k_{RISC}$, as SOC generally affects the whole spin-flip processes, not only $k_{RISC}$ but also $k_{ISC}$. Consequently, the ISC process occurs much more rapidly than the RISC process in this molecular system, which should interrupt conquest in regard to an ideal exciton lifetime. Therefore, the balance of rate constants $k_r$, $k_{ISC}$, and $k_{RISC}$ is the decisive factor controlling the exciton lifetimes. In other words, a larger $k_r$ than $k_{ISC}$ and minimal difference between $k_{ISC}$ and $k_{RISC}$ while maintaining $^1$CT ≈ $^3$CT ≈ $^3$LE are concurrently required for TADF molecules possessing ideally fastest delayed lifetime.

In summary, we successfully designed and synthesized an advanced deep-blue TADF emitter, **TMCz-BO**, which displayed deep-blue emission with corresponding CIE chromaticity coordinates of (0.15, 0.18) and (0.14, 0.18) under photo- and electro-excitation, respectively. Owing to its extraordinarily fast emission lifetime of 750 ns, originating from well-valanced $k_r$, $k_{ISC}$, and $k_{RISC}$ mediated by suitably strong SOC between the $^1$CT and $^3$LE while maintaining a small activation energy, high external EL quantum efficiencies of 20.2% and 17.4% at 100 cd m$^{-2}$ for displays and 1000 cd m$^{-2}$ for lighting sources were achieved, respectively. We presume that our strategy will be widely employed in TADF molecular design for high-performance deep-blue TADF-OLEDs.

## Methods

**General**. All reagents and anhydrous solvents were purchased from commercial sources and were used without further purification. The detailed synthetic procedures and characterization data for the intermediates are given in Supplementary Figs. 8–11. The two final products investigated in this paper were synthesized by following the procedures described below and then purified using temperature-gradient vacuum sublimation with a P-100 system (ALS Technology). $^1$H and $^{13}$C NMR spectra were recorded on a Bruker Avance III 500 spectrometer. Chemical shifts of $^1$H and $^{13}$C NMR signals were quoted to tetramethylsilane ($\delta = 0.00$) and CDCl$_3$ ($\delta = 77.0$), as internal standards. Mass spectra were measured in positive-ion atmospheric-pressure chemical ionization mode on a Waters 3100 mass detector. Elemental analyses were performed using a Yanaco MT-5 analyzer. For thin films and OLED fabrication, CCP[16] and PPF[44] were synthesized following the reported procedures and were then purified using temperature-gradient vacuum sublimation. Other OLED materials were purchased from Luminescence Technology Corporation.

**Synthesis of TMCz-BO**. A solution of *n*-BuLi in hexane (3.11 mL, 2.5 M, 7.77 mmol) was added slowly to a solution of **2** (3.0 g, 6.47 mmol) in *tert*-butylbenzene (30 mL) at −20 °C under a nitrogen atmosphere. After stirring at 50 °C for 4 h, hexane was distilled off at 100 °C under a continuous flow of nitrogen. BBr$_3$ in dichloromethane (9.71 mL, 1.0 M, 9.71 mmol) was added slowly at −20 °C. The reaction mixture was then allowed to warm to room temperature for 1 h and then stirred at 40 °C for 1 h. After 10% of the solvent had been removed in vacuo, N,N-diisopropylethylamine (DIPEA, 1.67 g, 12.9 mmol) was added at 0 °C. After stirring

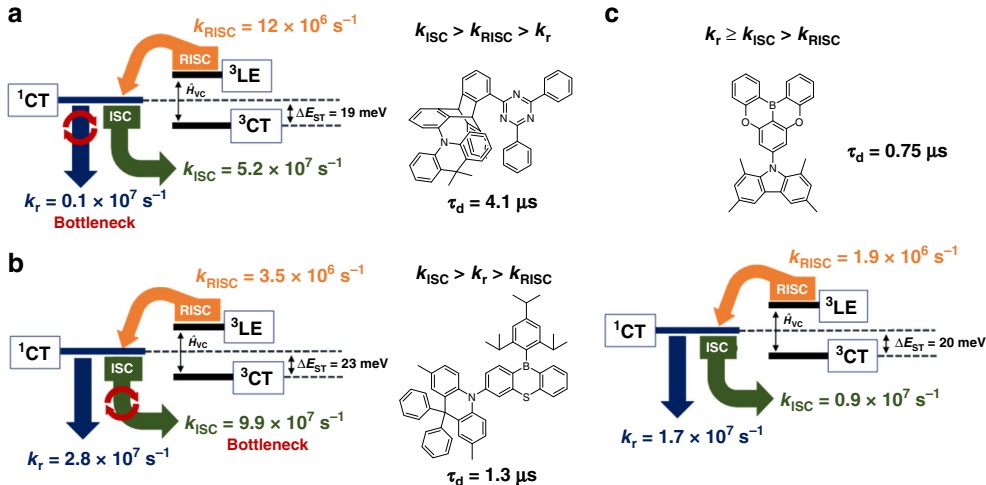

**Fig. 7 Schematic illustration of plausible TADF mechanism. a** TpAT-tFFO, **b** MPAc-BS, and **c** TMCz-BO. $k_r$, $k_{ISC}$, $k_{RISC}$, CT, LE, and $\Delta E_{ST}$ represent the rate constants of fluorescence radiative decay, intersystem crossing (ISC), reverse ISC (RISC), charge-transfer and localized excited states, and energy gap between lowest singlet and triplet excited states, respectively.

at 120 °C for 5 h, methanol was added to the reaction mixture at room temperature. The reaction mixture was then filtered with a pad of Florisil. The crude product was washed with methanol to afford **TMCz-BO** (0.48 g, 15% yield) as a light-yellow solid. $^1$H NMR (500 MHz, CDCl$_3$, $\delta$): 8.77 (dd, $J = 7.8$, 1.5 Hz, 2H), 7.77 (td, $J = 8.1$, 1.5 Hz, 4H), 7.58 (dd, $J = 7.8$, 0.6 Hz, 2H), 7.46 (td, $J = 7.4$ Hz, 0.9 Hz, 2H), 7.40 (s, 2H), 6.95 (s, 2H), 2.51 (s, 6H), 1.97 (s, 6H) (Supplementary Fig. 12); $^{13}$C NMR (125 MHz, CDCl$_3$, $\delta$): 160.65, 156.88, 148.61, 139.60, 134.66, 133.99, 130.29, 129.30 124.42, 123.22, 121.44, 118.63, 117.79, 112.01, 21.11, 19.33. (Supplementary Fig. 13); MS (ASAP) $m/z$: $[M-1]^+$ calcd 493.39; found, 492.17. Anal. calcd (%) for C$_{34}$H$_{26}$BNO$_2$: C 83.10, H 5.33, N 2.85; found: C 83.09, H 5.36, N 2.85.

**Synthesis of TMCz-3P.** TMCz-3P was synthesized using the same procedure described above for the synthesis of **TMCz-BO**, except that **4** (3.0 g, 4.72 mmol), $n$-BuLi (2.26 mL, 5.66 mmol), BBr$_3$ (7.08 mL, 7.08 mmol), and DIPEA (1.22 g, 9.44 mmol) were used as the reactants, giving **TMCz-3P** (0.46 g, 15% yield) as a light-yellow solid. $^1$H NMR (500 MHz, CDCl$_3$, $\delta$): 8.74 (d, $J = 8.1$ Hz, 2H), 7.70 (td, $J = 7.6$, 1.4 Hz, 8H), 7.64 (dd, $J = 8.0$, 1.7 Hz, 2H), 7.46(t, $J = 7.6$ Hz, 4H), 7.38 (tt, $J = 7.35$, 3.0 Hz, 2H), 7.34 (s, 2H), 6.90 (s, 2H), 6.87 (s, 2H), 2.42 (s, 6H), 1.90 (s, 6H) (Supplementary Fig. 14); $^{13}$C NMR (125 MHz, CDCl$_3$, $\delta$): 160.07, 155.98, 147.48, 145.79, 138.52, 134.04, 129.28, 128.26 128.01, 127.36, 126.34, 123.37, 120.39, 116.78, 115.58, 111.10, 20.09, 18.33 (Supplementary Fig. 15); MS (ASAP) $m/z$: $[M]^+$ calcd 643.58; found, 643.23. Anal. calcd (%) for C$_{46}$H$_{34}$BNO$_2$: C 85.85, H 5.32, N 2.18; found: C 85.22, H 5.37, N 2.13.

**Quantum chemical calculations.** Quantum chemical calculations were performed using the Gaussian 16 program package. The molecular geometries in the ground state were optimized using the PBE0 functional with the 6-31G(d) basis set in the gas phase. The lowest excited singlet and triplet states as well as the TSDD simulation were computed using the optimized structures with TDDFT at the same level. The SOC matrix elements were determined using the ADF2018 program package[50] following the literature method[27].

**Photophysical measurements.** Thin-film samples (40 nm) were deposited on quartz glass substrates by vacuum evaporation to study their exciton confinement properties in the film state. UV–vis absorption and PL spectra were recorded using a PerkinElmer Lambda 950 KPA spectrophotometer and JASCO FP-6500 fluorescence spectrophotometer, respectively. The absolute PL quantum yields were measured on a Quantaurus-QY measurement system (C11347-11, Hamamatsu Photonics) under nitrogen flow, and all the samples were excited at 360 nm. The transient PL decay characteristics were recorded using a Quantaurus-Tau fluorescence lifetime measurement system (C11367-03, Hamamatsu Photonics). The HOMO energy levels were determined using the onset of a photoelectron yield spectroscopy (AC-3, Riken-Keiki) in neat films, and then the LUMO energy levels were estimated by subtracting the optical energy gap ($E_g$) from the measured HOMO energies. In the case of the $E_g$ values were determined from the onset of the PL spectra of neat films.

**Device fabrication and measurements.** Pre-patterned ITO (50 nm)-coated glass substrates were cleaned with detergent, deionized water, acetone, and isopropanol. The substrates were then exposed to UV–ozone treatment for 15 min before being loaded into an ALS Technology E-200 vacuum evaporation system. The organic

layers and cathode Al layer were thermally evaporated on the substrates under vacuum ($<6 \times 10^{-5}$ Pa) at a deposition rate of $<0.2$ nm s$^{-1}$ through a shadow mask, defining a pixel size of 0.04 cm$^2$. The thickness and deposition rate were monitored in situ during deposition by an oscillating quartz thickness monitor. The current density–voltage–luminance ($J$–$V$–$L$) characteristics of the fabricated OLEDs were measured using a Keithley 2400 source meter and a CS-2000 spectroradiometer (Konica Minolta).

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

## Acknowledgements

This work was supported by a grant from the Regional Innovation Eco-System Program sponsored by the Ministry of Education, Culture, Sports, Science and Technology (MEXT) of Japan, JST ERATO (Grant Number JPMJER1305), the International Institute for Carbon Neutral Energy Research (WPI-I2CNER) sponsored by MEXT, and JSPS KAKENHI (Grant Numbers 17H01232).

## Author contributions

C.A. initiated and supervised the project. J.U.K. designed, synthesized, and characterized the blue TADF emitters. J.U.K. performed the computational calculation, photophysical, and electrochemical measurements of the TADF emitters. J.U.K., I.S.P., and M.T. fabricated the OLEDs and measured the device performance. Y.T., H.N., and C.A. provided suggestions on experiments and writing manuscript. All authors discussed the progress of the research and reviewed the manuscript.

## Competing interests

The authors declare no competing interests.
