## [Peer Review File · Nature Communications]

Reviewers' comments:

Reviewer #1 (Remarks to the Author):

Comments

This manuscript describes molecularly engineered two donor-acceptor TADF emitters, which the authors incorporated into an OLED device architecture. The significance of the molecular design of the two emitters is overlap between the HOMO and LUMO that induce the CT character in both S1 and T1 states, resulting in a small $1CT \approx 3CT$ and fast exciton lifetime of 750 ns. The OLED devices containing TADF emitters exhibited low turn-on voltage (V_{on}) 2.8–3.0 V and achieved a high maximum efficiency of over 20%, which decreased with increasing current density and luminance. The new TADF emitter displayed deep-blue electroluminescence (EL) with CIE chromaticity coordinates of (0.14, 0.18), which is quite interesting therefore recommend for publication with minor revision.

(1). The introduction section is quite elaborate and informative but not essential. Therefore, the authors could consider reducing it significantly.

(2). The Figure 2b, table 1, why the emission maxima are red-shifted by 21 nm in 30% doped with 2,8-bis(diphenyl phosphoryl)dibenzo[b,d]furan films compared to toluene solution spectra?

Reviewer #2 (Remarks to the Author):

This article from Chihaya Adachi and coworkers reports structurally interesting donor-acceptor molecules which possess very short exciton lifetimes and efficient blue emission in OLEDs. These features are very important for current studies. This is a good molecular design and the work is rationally presented. Photophysical and device studies are presented for the emitters, combined with standard theoretical calculations. The SI provides sufficient information for other workers to repeat the synthesis and the structures of the compounds are appropriately confirmed, notably by figures of NMR spectra.

I am not qualified to assess the details of the theoretical calculations. My assessment is based on design, synthesis, experimental characterization and OLEDs. The work could become suitable for Nature Communications subject to the following revisions.

1. There is much emphasis placed in the manuscript on achieving the very short exciton lifetime, and the obtained 750 ns for TMCz-BO is impressive. However, it appears to be not essential for small roll-off in blue TADF OLEDs. The recent work of P. Stachelek et al, ACS Appl. Mater. Interfaces 2019, 11, 27125 reports a molecule with a significantly longer exciton lifetime that gives OLEDs with very similar CIE, EQE and small roll-off. The authors should cite and comment on this work in the manuscript and list the data in Table S3 in SI.

2. The solution electrochemical data should be shown for the new emitters TMCz-BO and TMCz-3P. This is standard characterization for a new D-A molecule. The experimental HOMO-LUMO energies should be compared with the computed values.

3. What percentage of the overall emission is from DF? Figures should be added to SI showing the oxygen dependence of the PL spectra.

4. Was PPF chosen as the host simply because of its triplet level. Or any other reason? Please comment.

5. A plot of device EQE versus brightness should be added to SI to clearly show the roll-off.

Reviewer #3 (Remarks to the Author):

Title: Nanosecond-Time-Scale Delayed Fluorescence Molecule for Deep-Blue Organic Light-Emitting Diodes with Small Efficiency Rolloff

In this manuscript, the authors reported a new deep-blue TADF emitter (TMCz-BO) having 1,3,6,8-tetramethyl-9H-carbazole (TMCz) as a donating unit and 5,9-dioxa-13b-boranaphtho[3,2,1-de]anthracene (BO) as a accepting unit. An extremely fast exciton lifetime of 750 ns was realized in a D-A type molecular structure without heavy metal elements. An OLED utilizing this new TADF emitter displayed deep-blue electroluminescence (EL) with CIE chromaticity coordinates of (0.14, 0.18) and a high maximum EL quantum efficiency of 20.7% with efficient roll-off behavior. I would like to recommend publishing after minor revisions. Comments of this paper are shown below:

(Q1) I would like to ask the authors to create one table containing the kinetic parameters of 30 wt% doped film in the support information. I think readers of the paper can understand it easily.

(Q2) In Table S1, I think the TMCz-3P's data is recorded incorrectly. I want the authors to confirm this again. (Eg, S1, T1).

(Q3) In Table S2, compared with TMCz-BO, I would like to ask the authors to briefly explain why the TMCz-3P in 30 wt% doped film has a larger E_a value.

(Q4) It would be nice to provide DSC and TGA for newly synthesized emitters.

(Q5) The structure of the TADF-OLED device and the structure of the molecules used in each layer would be shown in the supporting information for readers to understand better.

Reply for the comments

We sincerely thank the reviewers for their careful reading of our manuscript with the valuable comments. In response to the reviewer's comments, we carefully revised our manuscript. Our responses to the referee's comments are as follows:

Response for the Reviewer 1

1) The introduction section is quite elaborate and informative but not essential. Therefore, the authors could consider reducing it significantly.

[Reply]:

Thank you for your appropriate comment. We understand it and revised the introduction with the minimum description.

Revision of introduction section (Line 29-43)

Organic light-emitting diodes (OLEDs) have been commercialized in flat panel displays and solid-state lighting applications, and significant efforts are still devoted to enhancing OLED performance. In OLEDs, the most important parameter is the internal quantum efficiency (η_{int}), which is theoretically limited to 25% in traditional fluorescence-based OLEDs, as only singlet excitons can be harvested under electrical excitation.^{1,2} On the other hand, the utilization of phosphorescent emitters containing heavy metals such as Ir, Pt, Os and Au enhanced intersystem crossing by the strong spin-orbit coupling (SOC), and these phosphorescent emitters can harvest not only singlet excitons but also triplet excitons, leading to an ideal η_{int} of nearly 100% in OLEDs.^{3,4} However, the weak metal-ligand coordination bonds result in a limited device lifetime in blue OLEDs.⁵⁻⁷ As an alternative approach, highly efficient thermally activated delayed fluorescence (TADF)-based OLEDs have recently been realized using simple aromatic compounds as an emitter.⁸ In this system, triplet excitons are efficiently upconverted from a lowest triplet state (T_1) to the lowest excited singlet state (S_1) by a reverse intersystem crossing (RISC) process, governed by a small energy gap (ΔE_{ST}) between the S_1 and T_1 states, resulting in a maximum η_{int} of close to 100%. Eventually, high external quantum efficiencies (η_{ext}) of over 20% have been achieved for TADF-OLEDs.⁹⁻³¹

2) The Figure 2b, table 1, why the emission maxima are red-shifted by 21 nm in 30% doped with 2,8-bis(diphenyl phosphoryl)dibenzo[b,d]furan films compared to toluene solution spectra?

[Reply]: Thank you for your comment. In the previous manuscript, we mention about the solvatochromism (lines 127-132 in the revised manuscript). **TMCz-BO** and **TMCz-3P** possess a CT character, meaning that they should be strongly influenced by the polarity of host matrices. As a result, the redshifted emission maxima of **TMCz-BO** and **TMCz-3P** in the PPF host matrix compared with those in toluene solution were observed. We note that the PPF host

possesses a stronger dipole moment ($\mu = 5.8$ D, *Adv. Funct. Mater.* **2018**, 28, 1802031) than that of toluene (0.36 D). Therefore, this phenomenon is an additional evidence that the molecules have a strong CT character. We put the extra explanation in the revised manuscript (lines 138-142).

Insertion of additional explanation (Line 139-143)

We note that the red-shifted emission maxima of the doped films were observed compared to those of their toluene solution ($\lambda_{\text{PL}} = 446$ nm for **TMCz-BO** and $\lambda_{\text{PL}} = 455$ nm for **TMCz-3P**), which should be primarily associated with a stronger dipole moment of PPF (5.8 D) than that of toluene (0.36 D).²⁷ Such a phenomenon is an additional evidence that these molecules possess a strong CT character.

Response for the Reviewer 2

1) There is much emphasis placed in the manuscript on achieving the very short exciton lifetime, and the obtained 750 ns for **TMCz-BO** is impressive. However, it appears to be not essential for small roll-off in blue TADF OLEDs. The recent work of P. Stachelek et al, *ACS Appl. Mater. Interfaces* 2019, 11, 27125 reports a molecule with a significantly longer exciton lifetime that gives OLEDs with very similar CIE, EQE and small roll-off. The authors should cite and comment on this work in the manuscript and list the data in Table S3 in SI.

[Reply] Thank you for proving this important update information. We cited this reference as 49 in the Device characterization and performance section and add the description. Also, we put the data in Table S4 in SI.

Insertion of sentences in Device characterization and performance section (Line 226-230)

We note that the recently some sophisticated devices showed rather relaxed rolloff behavior even with the TADF emitters with microsecond order delayed lifetimes. Probably wide recombination and exciton formation width may contribute it. Thus, it should be considered to improve total device performance from the aspect of comprehensive control of delayed lifetime, charge carrier transport and recombination characteristics.⁴⁹

2) The solution electrochemical data should be shown for the new emitters **TMCz-BO** and **TMCz-3P**. This is standard characterization for a new D-A molecule. The experimental HOMO-LUMO energies should be compared with the computed values.

[Reply] Thank you for this comment. We agree to put HOMO and LUMO information. In our study, the HOMOs were determined using the onset of a photoelectron yield spectroscopy (AC-3, Riken-Keiki) in their neat films, and the LUMOs were estimated by subtracting the optical energy gap (E_g) from the measured HOMOs. Here, E_g values were determined from the onset of the absorption spectra in Tol solution (10^{-5} M). These experimental results and

explanation were added in the revised manuscript (Table 1 and Methods Section) and Supplementary Information (Figure S9), respectively.

Figure S9. (a) Photoelectron yield spectra (in films) and (b) absorption spectra (in Tol solution 10⁻⁵ M) of TMCz-BO and TMCz-3P.

3) What percentage of the overall emission is from DF? Figures should be added to SI showing the oxygen dependence of the PL spectra.

[Reply] Thank you for your comment. While we have already mentioned the percentage of DF in the original manuscript (lines 144-154), we further measured the oxygen dependence of PL spectra/intensity in toluene solution (10⁻⁵ M) and doped films (30 wt% in PPF). These experimental results were added in the revised Supplementary Information (Fig. S7).

Figure S7. Oxygen dependence of PL spectra for TMCz-BO and TMCz-3P in (a) toluene solution (10⁻⁵ M), and (b) 30 wt% doped films in a PPF host matrix.

4) Was PPF chosen as the host simply because of its triplet level. Or any other reason? Please comment.

[Reply]: The main reason of employing PPF as the host is its high T_1 energy (3.1 eV) to maintain a highly triplet confinement in the host-guest system. In addition, PPF has a deeper LUMO energy level (-2.7 eV) than that of most widely used DPEPO (-2.0 eV). Thus, PPF should induce more efficient electron-injection and -transfer in our devices. We added the description.

Insertion of sentences, Line 135-136

Here, we selected PPF as a host since it can provide both efficient triplet confinement and electron transport abilities.

5) A plot of device EQE versus brightness should be added to SI to clearly show the roll-off

[Reply]: We updated a plot of EQE versus luminance for the devices in the revised Supplementary Information (Fig. S11).

Figure S11. (a) Energy level diagram and (b) external EL quantum efficiency (η_{ext}) versus luminance plots of the OLEDs with TMCz-BO and TMCz-3P. (c) Molecular structures of materials used for TADF-OLEDs.

Response for the Reviewer 3

1) We would like to ask the authors to create one table containing the kinetic parameters of 30 wt% doped film in the support information. I think readers of the paper can understand it easily.

[Reply] We update and add some photophysical parameters including HOMOs and LUMOs in **Table 1** and also prepared new **Table S2** that summarized the kinetic parameters.

Table 1. Photophysical characteristics of **TMCz-BO** and **TMCz-3P**.

Compound	λ_{abs}	λ_{PL}	$\Phi_{\text{PL}}[\%]^{\text{c}}$	τ_{p}	HOMO	LUMO	$E_{\text{S}} / E_{\text{T}}$	ΔE_{ST}	E_{a}
	[nm]	[nm]		[ns] ^d					
	sol ^a	sol ^a / film ^b	sol ^a / film ^b	τ_{d}					
				[μs] ^d					
TMCz-BO	282,377	446 / 467	81 / 98	38 / 0.75	-5.93	-2.95	2.95 / 2.93	20.0	13.4
TMCz-3P	298,386	455 / 477	56 / 76	29 / 14.5	-5.97	-3.23	2.88 / 2.74	134	39.8

^a) Measured in oxygen-free toluene at room temperature (298 K); ^b) 30 wt% doped thin film in a host matrix (host = PPF); ^c) Absolute PLQY evaluated using an integrating sphere under a nitrogen atmosphere; ^d) PL lifetimes of prompt (τ_{p}) and delayed (τ_{d}) decay components for 30 wt% doped film at 298 K; ^e) Determined by photoelectron yield spectroscopy in pure neat films; ^f) Deduced from the HOMO and optical energy gap (E_{g}); ^g) Singlet (E_{S}) and triplet (E_{T}) energies estimated from onsets of the emission spectra at 298 K and 77 K in 30 wt% doped films, respectively; ^h) $\Delta E_{\text{ST}} = E_{\text{S}} - E_{\text{T}}$; ⁱ) Activation energies of RISC in 30 wt% doped films.

Table S2. Rate constants and quantum efficiencies of **TMCz-BO** and **TMCz-3P** in 30 wt%-doped films^a

Compounds	k_{r}	k_{ISC}	k_{RISC}	Φ_{p}	Φ_{d}	Φ_{ISC}	Φ_{RISC}
	[s ⁻¹]	[s ⁻¹]	[s ⁻¹]	[%]	[%]	[%]	[%]
TMCz-BO	1.7×10^7	0.9×10^7	1.9×10^6	66	32	34	96
TMCz-3P	2.3×10^7	1.2×10^7	3.3×10^4	65	11	35	41

^a) Abbreviations: k_{r} , radiative rate constant ($\text{S}_1 \rightarrow \text{S}_0$); k_{ISC} , intersystem-crossing (ISC) rate constant ($\text{S}_1 \rightarrow \text{T}_1$); k_{RISC} , reverse ISC rate constant ($\text{T}_1 \rightarrow \text{S}_1$); Φ_{p} , quantum efficiency for prompt fluorescence component; Φ_{d} , quantum efficiency for delayed fluorescence component; Φ_{ISC} , ISC quantum efficiency; Φ_{RISC} , RISC quantum efficiency.

2) In Table S1, I think the TMCz-3P's data is recorded incorrectly. I want the authors to confirm this again. (Eg, S1, T1).

[Reply] We are sorry for this mistake. The error was fixed in the revised Supplementary Information (Table S1).

Table S1. Summary of TDDFT calculations (PBE0/6-31G(d)).

Compound	HOMO [eV]	LUMO [eV]	E_g [eV]	E_S / E_T [eV]	ΔE_{ST} [eV]	$f@S_0$	μ [D]	λ_s [eV]
TMCz-BO	-5.29	-1.83	3.46	2.56 / 2.55	0.01	0.0057	3.39	0.15
TMCz-3P	-5.28	-1.93	3.35	2.47 / 2.46	0.01	0.0013	3.83	0.16

3) In Table S2, compared with TMCz-BO, I would like to ask the authors to briefly explain why the TMCz-3P in 30 wt% doped film has a larger E_a value.

[Reply] Thank you for your insightful comment. Basically, it is supposed that E_a should be associated with the positions of 3CT and 3LE (donor and acceptor) energy levels in TMCz-3P. However, we could not provide an appropriate answer on this phenomenon at this stage. In our subsequent work, we would like to clarify this phenomenon by analyzing the detailed energy structures of 3LE and 3CT levels.

4) It would be nice to provide DSC and TGA for newly synthesized emitters.

[Reply] Thank you for your comment. We analyzed thermal properties of **TMCz-BO** and **TMCz-3P** using DSC and TGA, which were recorded at a heating rate of $5\text{ }^\circ\text{C min}^{-1}$ under N_2 atmosphere. These experimental results were added in the revised Supplementary Information (Fig. S10).

Figure S10. TGA and DSC (inset) curves for **TMCz-BO** and **TMCz-3P** recorded at a heating rate of $5\text{ }^{\circ}\text{C min}^{-1}$ under N_2 . T_d is the decomposition temperature, corresponding to 5% weight loss upon heating and T_g is the glass-transition temperature.

5) The structure of the TADF-OLED device and the structure of the molecules used in each layer would be shown in the supporting information for readers to understand better.

[Reply] We updated the structure of the TADF-OLEDs with the molecules used for each layer in the revised Supplementary Information (Fig. S11).

Figure S11. (a) Energy level diagram and (b) external EL quantum efficiency (η_{ext}) versus luminance plots of the OLEDs with **TMCz-BO** and **TMCz-3P**. (c) Molecular structures of materials used for TADF-OLEDs.

REVIEWERS' COMMENTS:

Reviewer #1 (Remarks to the Author):

The authors have improved the manuscript by taking into account the referee's comments. The revisions are satisfactory, and the manuscript is acceptable for publication.

Reviewer #2 (Remarks to the Author):

The authors have satisfactorily revised the manuscript in accord with the reviewers' comments and questions. In my opinion the manuscript can be accepted without further changes.

Reviewer #3 (Remarks to the Author):

I have read their rebuttal letter and their revised manuscript.
The authors have revised the manuscript according to the review comments properly. I think it is suitable to be published in nature communications at the current revised manuscript.